# Hidden structural and chemical order controls lithium transport in cation-disordered oxides for rechargeable batteries

Huiwen Ji[1,2], Alexander Urban[1,2,3], Daniil A. Kitchaev[4], Deok-Hwang Kwon[1,2], Nongnuch Artrith [1,2], Colin Ophus[5], Wenxuan Huang[4], Zijian Cai[1,2], Tan Shi[1,2], Jae Chul Kim[2,6], Haegyeom Kim[2] & Gerbrand Ceder[1,2]

Structure plays a vital role in determining materials properties. In lithium ion cathode materials, the crystal structure defines the dimensionality and connectivity of interstitial sites, thus determining lithium ion diffusion kinetics. In most conventional cathode materials that are well-ordered, the average structure as seen in diffraction dictates the lithium ion diffusion pathways. Here, we show that this is not the case in a class of recently discovered high-capacity lithium-excess rocksalts. An average structure picture is no longer satisfactory to understand the performance of such disordered materials. Cation short-range order, hidden in diffraction, is not only ubiquitous in these long-range disordered materials, but fully controls the local and macroscopic environments for lithium ion transport. Our discovery identifies a crucial property that has previously been overlooked and provides guidelines for designing and engineering cation-disordered cathode materials.

[1] Department of Materials Science and Engineering, University of California Berkeley, Berkeley, CA 94720, USA. [2] Materials Sciences Division, Lawrence Berkeley National Laboratory, Berkeley, CA 94720, USA. [3] School of Chemistry, University of St Andrews, St Andrews KY16 9ST, UK. [4] Department of Materials Science and Engineering, Massachusetts Institute of Technology, Cambridge, MA 02139, USA. [5] National Center for Electron Microscopy, Molecular Foundry, Lawrence Berkeley National Laboratory, Berkeley, CA 94720, USA. [6] Department of Chemical Engineering and Materials Science, Stevens Institute of Technology, Hoboken, NJ 07030, USA. Correspondence and requests for materials should be addressed to G.C. (email: gceder@berkeley.edu)

The development of cost-effective lithium (Li) ion batteries depends on the discovery of high-energy-density cathode materials composed of non-precious elements[1]. Rational design of cathodes requires an understanding of the precise role that each chemical component has in determining performance. Traditionally, one thinks of redox-active elements, such as Ni and Co, and stabilizers, such as $Mn^{4+}$ in NMC-class materials[2]. However, we demonstrate in this paper that in the recently discovered class of Li-excess cation-disordered rocksalt (DRX) cathodes, the chemistry of then non-redox-active stabilizers plays a critical role in performance through subtle structural changes.

DRX materials were recently shown to have facile Li transport enabled by a percolating network of Li-rich environments[3]. Their ability to function without requiring cation ordering has enabled novel cathodes with remarkable chemical diversity[3,4]. Many new cathode materials, in some cases containing only earth-abundant elements (e.g., Fe, Mn, and Ti) have been developed in this category, such as $Li_{1.3}Mn_{0.4}Nb_{0.3}O_2$[5], $Li_{1.2}Mn_{0.4}Ti_{0.4}O_2$[6], $Li_{1.2}Ni_{1/3}Ti_{1/3}Mo_{2/15}O_2$[7], $Li_4Mn_2O_5$[8], and $Li_2FeV_{0.5}Ti_{0.5}O_4$[9] as well as their fluorinated variants[10–14]. A prevailing assumption when studying DRX cathodes is that all the cation species are randomly distributed.

Figure 1a presents a typical DRX crystal structure. The highlighted tetrahedron represents a channel through which Li migrates[15] (Fig. 1b). The Li-migration barrier depends on the tetrahedron height[3,16–18] and the number of transition metal (TM) ions within the environment (i.e. 0-TM, 1-TM, or 2-TM)[4]. If the cation arrangement is random, any DRX with the same Li to TM ratio should have an equivalent distribution of Li-migration channel types and thus similar Li transport properties.

However, in this study, we show that even minor deviations from randomness, not detectable by typical X-ray diffraction (XRD), can have profound influence on performance. We find that cation short-range order (SRO) is important in determining the amount of kinetically extractable Li in DRX materials as the Li transport relies on percolation through a 3D network. We

compare two Li-excess DRXs, $Li_{1.2}Mn_{0.4}Ti_{0.4}O_2$ (LMTO) and $Li_{1.2}Mn_{0.4}Zr_{0.4}O_2$ (LMZO). Based on their chemical similarity, these materials would be expected to have comparable electrochemical properties, as $Zr^{4+}$ and $Ti^{4+}$ are isoelectronic and their sole role is to charge compensate for the excess Li. If anything, the larger ionic radius of $Zr^{4+}$ should result in a larger lattice parameter for LMZO, which is generally considered beneficial for Li mobility[16–18]. However, contrary to these expectations, we observe that the performance of LMTO is considerably better than that of LMZO. We reveal through a combination of electron diffraction, neutron pair distribution function measurements, and cluster-expansion Monte Carlo simulation that the difference in the performance of LMTO and LMZO is due to different cation SRO, which controls the population and connectivity of Li-migration channels. We further identify general rules that govern the relationship between SRO and Li transport by expanding our analysis to other combinations of TMs. These results indicate the importance of SRO and provide another important handle to tailor the performance of DRX cathode materials, in addition to the already large compositional flexibility.

## Results

**Synthesis and electrochemistry.** We synthesized LMTO and LMZO using a solid-state method and verified the compositions of the products to be nearly identical to the target compositions using elemental analysis (Supplementary Table 1 and Supplementary Note 1). XRD patterns (Fig. 1c) reveal a DRX structure. Rietveld refinement indicates that the lattice parameter of LMZO ($a = 4.27$ Å) is larger than that of LMTO ($a = 4.15$ Å), as expected (Supplementary Figure 1). Scanning electron microscopy (SEM) images of shaker-milled LMTO and LMZO (Fig. 1d and e) confirm particle sizes of ~100 nm for both materials. The galvanostatic voltage profiles of LMZO and LMTO are presented in Fig. 1f and g. Consistent with a previous report by Yabuuchi et al.[6], LMTO delivers a large first-cycle capacity of ~260 mAh g$^{-1}$ at

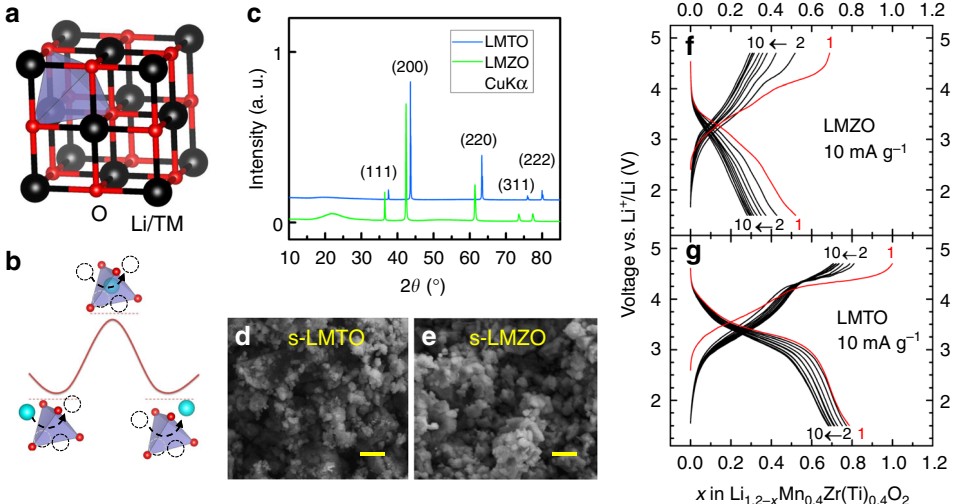

**Fig. 1** Characterization and electrochemical performance of $Li_{1.2}Mn_{0.4}Zr_{0.4}O_2$ (LMZO) and $Li_{1.2}Mn_{0.4}Ti_{0.4}O_2$ (LMTO). **a** Crystal structure of an ideal cation-disordered rocksalt-type lithium metal oxide. The black spheres represent metal cations (including lithium and TMs), and the red spheres represent oxygen anions. Both cations and anions are in octahedral coordination. The highlighted blue tetrahedral site represents a typical migration pathway for Li diffusion. **b** Schematic energy landscape of Li migration from its octahedral coordination through a tetrahedral vacancy into another octahedron. The energy barrier depends on the local environment and size of the tetrahedron. The migrating Li ion is highlighted in cyan. **c** XRD patterns of LMZO and LMTO indexed according to the rocksalt structure. The low-angle shift in the pattern of LMZO compared with that of LMTO indicates the larger lattice parameter of LMZO. **d, e** SEM images (scale bars, 500 nm) of shaker-milled LMTO (s-LMTO) and LMZO (s-LMZO) with similar particle sizes of ~100 nm. **f, g** Voltage profiles of LMZO and LMTO between 1.5 and 4.7 V for the first 10 cycles at room temperature

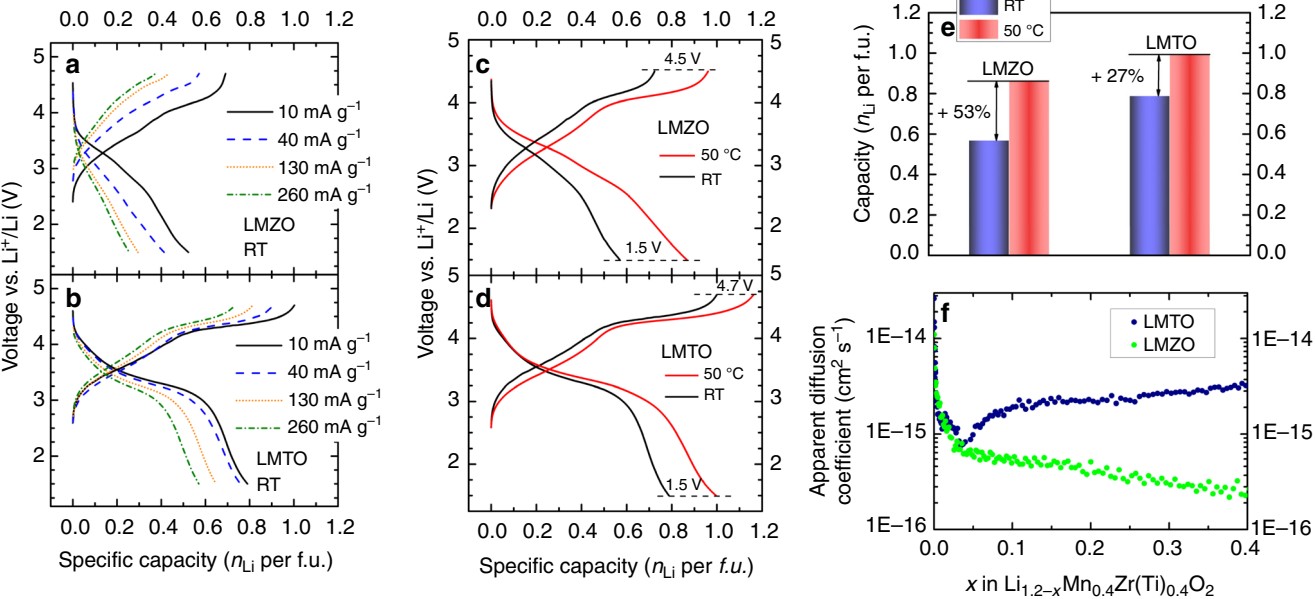

**Fig. 2** Rate capability tests, comparison between high-temperature and room-temperature galvanostatic cycling of LMTO and LMZO, and PITT measurements. First-cycle voltage profiles of LMZO (**a**) and LMTO (**b**) when cycled between 1.5 and 4.7 V at 10, 40, 130, and 260 mA g⁻¹. First-cycle galvanostatic voltage profiles of **c** LMZO and **d** LMTO at 50 °C and room temperature at 10 mA g⁻¹. **e** First-cycle reversible capacities of LMZO and LMTO at high temperature and room temperature. **f** Li chemical diffusion coefficients of LMZO (green) and LMTO (blue) obtained from fitting the room-temperature PITT data at various Li contents

room temperature, which corresponds to 0.79 Li per formula unit (*f.u.*). However, LMZO delivers a much smaller discharge capacity (0.52 Li per *f.u.*) (ex situ XRD patterns and supplementary electrochemical tests are shown in Supplementary Figures 2, 3 and described in Supplementary Note 2). In addition, the average discharge voltage for the first cycle is lower in LMZO (2.75 V) than in LMTO (3.08 V).

The Li diffusion is a key factor in determining the observed capacity[19,20], especially in LMTO and LMZO. The computed voltage profiles obtained by enumerating a large number of possible configurations (Supplementary Figure 4 and Supplementary Note 3) indicate that both materials should cycle in a comparable voltage window and their capacity would not be limited by thermodynamics. Indeed, the rate capability of LMTO (Fig. 2b) is clearly better than that of LMZO (Fig. 2a). To further test whether the capacity of LMZO is limited by kinetics, galvanostatic cycling at both room-temperature and 50 °C was performed, as shown in Fig. 2c, d. The upper cutoff voltage for LMZO is set at 4.5 V, while that of LMTO is set at 4.7 V. This is because we find that cycling LMZO to 4.7 V at 50 °C provides little extra reversible capacity compared to a 4.5 V-cutoff and causes a side reaction above 4.5 V which seems to lead to increased polarization and subsequent discharge voltage fade. The difference between room-temperature and 50 °C indicates pronounced kinetic limitations in LMZO over the entire voltage range, unlike LMTO, which only shows improvement at 50 °C near the upper cutoff voltage. As shown in Fig. 2e, at 50 °C, LMZO delivers a reversible capacity of 0.87 Li per *f.u.*, a 53% increase from that at room temperature. In contrast, the capacity of LMTO improves by only 27% to 1.0 Li per *f.u.*, when cycled at 50 °C. The Li chemical diffusivities ($D_{Li}$) of LMTO and LMZO were determined using the potentiostatic intermittent titration technique (PITT)[21–23] during the initial charge from open-circuit voltages to 4.7 V (Fig. 2f, Supplementary Figure 5 and Supplementary Note 4). It should be noted that the apparent Li diffusivity deduced from PITT includes the contribution from bulk and grain boundaries whereas the migration barriers from

nudged elastic band calculations would only impact the bulk diffusivity. Besides, given that the chemical diffusivity measured by PITT is a product of the thermodynamic factor and the self-diffusivity, the apparent diffusivities ($D_{Li}$'s) of the two materials at the beginning of charge ($x < 0.05$) are dominated by the large thermodynamic factors originating from the steep voltage increase in the region and are therefore not a representation of the intrinsic Li mobility. We observe that the Li diffusivity in LMTO is much higher than that in LMZO, confirming that the capacity of LMZO is more limited by Li transport kinetics. Given the validity of the PITT method in solid-solution systems, we plot the region from $x = 0$ to $x = 0.4$ where $Mn^{3+/4+}$ oxidation and a solid-solution reaction dominate[6,24].

**Local-structure characterization**. The discovery that two almost identical materials exhibit significantly different Li transport led us to investigate the subtle structural differences between LMTO and LMZO. Figure 3a, c and Supplementary Figure 6 present the electron diffraction (ED) patterns of LMTO and LMZO. Aside from the reflection spots that can be indexed with a DRX structure, we also observe diffuse scattering patterns surrounding the Bragg reflections, suggesting the existence of SRO. The formation of SRO can be understood as a preferred local arrangement of species, resulting in non-vanishing patterns in reciprocal space[25–28]. Notably, the diffuse scattering patterns are completely different, both in shape and orientation, for the two materials, indicating significant difference in SRO. Based on previous characterization of SRO in oxides and alloys, the SRO patterns in LMTO is characteristic of octahedral cation clusters similar to the [Li₃Fe₃] in cubic-LiFeO₂[29], whereas that in LMZO is likely associated with tetrahedral cation clusters[30]. In addition, the intensity of the diffuse scattering pattern of LMZO is noticeably stronger at several maxima highlighted with yellow arrows, suggesting more pronounced SRO in LMZO than in LMTO. These intensity maxima at $\langle 1\frac{1}{2}0 \rangle$ positions are among the four classes of special points in an FCC lattice where more than two symmetry elements intersect and do not correspond to any

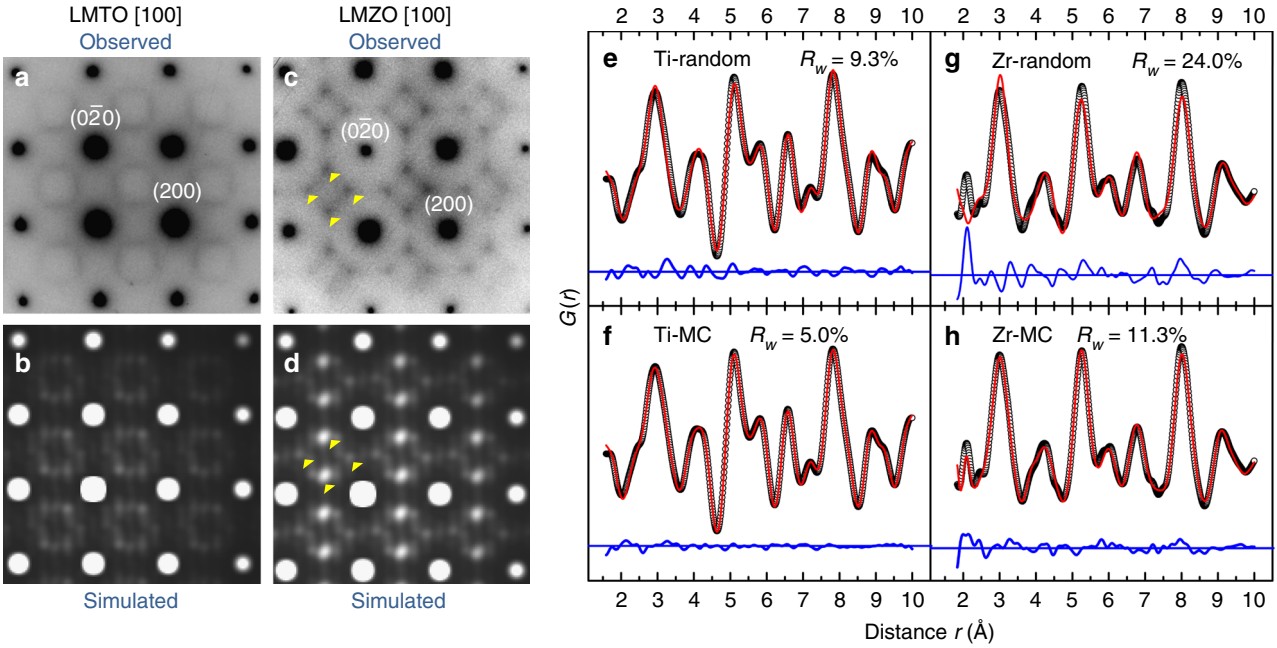

**Fig. 3** Experimental observation and computational simulation of short-range order (SRO) in LMTO and LMZO. ED patterns of LMTO (**a**) and LMZO (**c**) along the zone axis [100]. The round spots are indexed to the $Fm\text{-}3m$ space group, while the diffuse scattering patterns nearby are attributed to SRO. Several intensity maxima in the diffuse scattering patterns are highlighted with yellow arrows in LMZO. Simulation of ED patterns for LMTO (**b**) and LMZO (**d**) along the same zone axis shows good agreement with experimental observation. Refinement of NPDF data of LMTO (**e**, **f**) and LMZO (**g**, **h**) using the random model (**e**, **g**) and MC-equilibrated structural models (**f**, **h**). The experimental data are plotted as black open circles. The calculated values are plotted as solid red lines. The difference between observation and calculation is plotted as solid blue lines

long-range order[31]. These characteristic features observed in the diffuse scattering patterns are well reproduced by simulation (Fig. 3b, d) based on thermodynamically representative structures, which we obtain from Monte Carlo (MC) sampling at 1000 °C with a cluster expansion (CE) Hamiltonian parameterized to fit the rocksalt configurational energies derived from density-functional theory (DFT)[32,33]. Detailed analysis of the model structures is presented in the next section.

Neutron pair distribution function (NPDF) measurements were performed to precisely characterize the SRO in LMTO and LMZO. In NPDF analysis, Fourier transformation of the total scattering data to real space is performed, thereby providing additional information about SRO that is hidden in diffraction patterns[34]. The refinement of the NPDF patterns of LMTO and LMZO is presented in Fig. 3e–h. We use two structural models for the refinement: One is a random model that assumes a random cation distribution in a distortion-free lattice, and the other is the MC-derived structural model described above. The random model produces a reasonable fit for LMTO but not for LMZO, as indicated by the goodness-of-fit values ($R_w$) (Fig. 3e, g, Supplementary Figure 9, Supplementary Note 5 and Supplementary Table 2). In LMZO, the simulation based on the random model differs significantly from the experimental observation near 2, 3, 5, and 8 Å (Fig. 3g). Nevertheless, in the longer range, the random model produces a good fit for both materials (Supplementary Figure 7 and Supplementary Table 3). These results suggest that LMZO has significantly more SRO than LMTO. As a comparison, the refinement using the MC configurations, presented in Fig. 3f, h, shows significant improvement for both compounds (details in Supplementary Figure 8, Supplementary Table 4 and Supplementary Method 1).

Combining the analysis of ED and NPDF, we find that LMTO and LMZO differ in their SRO and that the ab initio MC

structures simulated near the synthesis temperature are precise manifestations of the SRO in these materials.

**Modeling of Li transport environments in LMTO and LMZO.** The structures obtained from MC simulation uncover the atomistic nature of the SRO, enabling further analysis of the local and macroscopic Li-transport environments in LMTO and LMZO.

In DRX materials, local environments can be characterized by the occurrence of cation clusters, among which the ones most relevant to Li transport are tetrahedral clusters, i.e., Li-migration channels[19]. Because of their connectivity in the structure, their population is not completely independent, a phenomenon that has been recognized early on when studying the entropy of FCC systems[35]. Figure 4a summarizes the occurrence of various tetrahedral clusters in LMTO and LMZO relative to a random case. We observe that the occurrence of $Li_4$ tetrahedra (i.e. 0-TM channels), which is the most important for good Li transport, is significantly lower in LMZO than in LMTO, although both materials have lower $Li_4$ population than for a random cation distribution. Conversely, the population of $Li_3M$ tetrahedra, i.e. 1-TM channels, is much higher in LMZO than in LMTO. More specifically, the $Li_3Zr$ clusters account for 31% of all cation tetrahedra in LMZO, as compared to 22% for $Li_3Ti$ in LMTO and 17% for $Li_3M'$ in the random case (Supplementary Table 5). Such a high population of 1-TM channels in LMZO is detrimental for Li transport as it was previously demonstrated that in a typical DRX, the migration barrier through a 1-TM channel is on average 200 meV higher than that through a 0-TM channel[3].

These observations indicate that SRO strongly modifies the population of local tetrahedral clusters: LMTO favors $Li_4$ clusters more than LMZO does; while LMZO contains more $Li_3M$ (especially $Li_3Zr$) clusters.

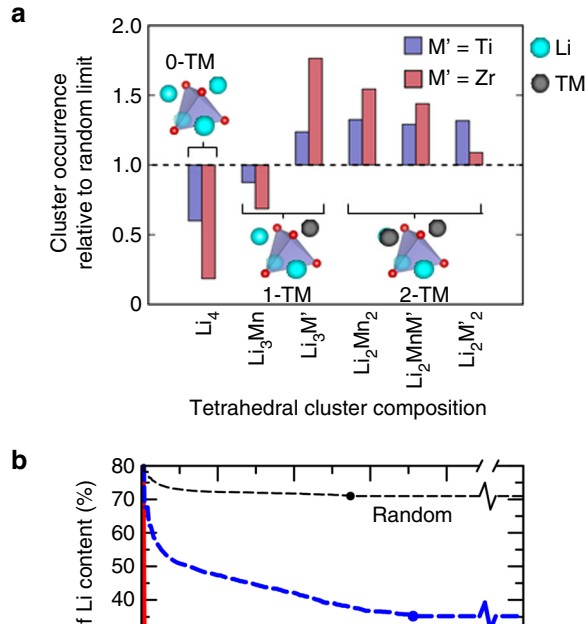

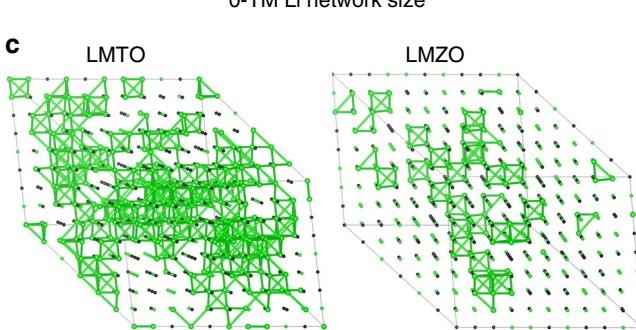

**Fig. 4** Local cation cluster and macroscopic Li connectivity analysis based on MC-derived structures for LMTO and LMZO at 1000 °C. Each MC structure contains 480 cation sites, of which 288 are decorated with Li ions. **a** Occurrence of various tetrahedral clusters (0-TM, 1-TM, 2-TM) in LMTO (blue) and LMZO (red) as compared to the random limit. **b** Connectivity plots of LMZO and LMTO showing the fraction of Li content in networks of at least a certain number of Li sites. A Li network is defined as all the Li sites that are inter-connected through 0-TM channels. Each plot is averaged over 600 sampled MC structures. The result for a random cation distribution with the same Li to TM ratio is also plotted as a reference. The plots are truncated at the percolating Li contents (marked by dots) and extended to infinity for LMTO and the random case. LMZO is not percolating. **c** Representative MC structures for LMTO and LMZO. Li ions are labeled with green spheres and 0-TM connected Li sites are bridged with green bonds

While the population of $Li_4$ tetrahedra is critical for local Li migration, a sufficient connectivity between these environments is another key criterion to ensure macroscopic Li transport. Figure 4b presents a connectivity analysis for LMTO and LMZO as compared to a random cation distribution. A connectivity plot, averaged over 600 sampled MC structures, shows the fraction of Li content in Li networks of more than a certain number of 0-TM

inter-connected Li sites (Supplementary Method 2). The fraction of Li content initially decreases with an increasing network size and finally plateaus at the percolating Li level, which is considered the lower bound of accessible Li. Figure 4b suggests that LMTO has more extensively connected 0-TM Li networks compared to LMZO, although both materials show worse Li connectivity than the random case. Specifically, in LMTO, nearly 40% of Li is in 0-TM networks of more than 100 Li sites and ~35% of Li is percolating. In contrast, in LMZO, the fraction of Li in networks of more than 25 Li sites is already vanishingly small and the material is not percolating.

To visualize the Li diffusion pathways in LMTO and LMZO, representative MC structures are shown in Fig. 4c. The 0-TM connected Li networks are highlighted in green. From these images, it is clear that LMTO has extensive 0-TM Li networks that are well connected and should allow facile Li transport, whereas LMZO lacks 0-TM channels, thereby impeding Li diffusion.

We may draw some useful insight about the impact of SRO on the preferred cation configurations around tetrahedral sites based on the relationship between SRO and its corresponding ground states. This is because SRO is generally considered an "intermediate state" between a random state and an ordered low-temperature ground state, and therefore, it often resembles the low-temperature long-range order[30]. In DRX oxides with multiple TM species, our calculations indicate that phase separation into ground-state compounds is preferred at low temperatures. For example, LMZO is thermodynamically favored to decompose into $LiMnO_2$ and $Li_2ZrO_3$ at low temperatures according to our calculations. $Li_2ZrO_3$ is known to have a $\gamma$-$LiFeO_2$-like structure[36], in which most cation tetrahedra are made up out of 2Li and 2Fe ions, which gives the compound very poor Li percolation[4]. Indeed, not even one Li site in $Li_2ZrO_3$ is connected to other Li sites through 0-TM channels even at a Li-excess level of 33.3%. In contrast, $Li_2TiO_3$ adopts a $Li_2MnO_3$-like structure[37], which contains a large amount of tetrahedra with 1TM and 3Li as well as tetrahedra with 4Li, which makes every Li site connected with other Li sites. The SRO and percolation properties observed in the cation-disordered materials are reminiscent of the low-temperature ground states, which explain why LMTO favors the segregation of Li in tetrahedra while LMZO exhibits poor Li conductivity.

**Chemistry dependence of SRO and Li transport environments**. With the understanding of how SRO affects Li transport in LMZO and LMTO, we can investigate other combinations of TMs to determine how the SRO-affected Li transport environments vary with chemistry. Figure 5 shows the accessible Li contents based on the percolation theory[3] in a variety of $Li_{1.2}M'_aM''_bO_2$ compounds under two conditions: (i) allowing only 0-TM jumps or (ii) allowing any given Li to make a single 1-TM jump before reaching the 0-TM percolating network. The rationale behind the chosen conditions is that although sufficient 0-TM channels are required for macroscopic Li transport, in reality, on the atomic scale Li ions can occasionally overcome higher migration barriers through 1-TM channels on the time scale of battery charge and discharge. Besides, since $d^0$ TMs are known to promote cation disorder in Li-TM oxides[38], all the hypothetical compositions contain a redox-active TM as well as a $d^0$ TM.

Consistent with the connectivity analysis, LMTO has a high accessible Li content of 35% with only 0-TM jumps, which increases to nearly 58% by allowing 1-TM jumps, whereas LMZO is not percolating under either condition. The accessible Li contents for $Mn^{3+}$–$Nb^{5+}$ are worse than LMTO, but still

significantly better than those of LMZO, consistent with the good performance of the $Mn^{3+}$–$Nb^{5+}$ materials at elevated temperatures[5]. $V^{3+}$–$Nb^{5+}$ is quite similar to $Mn^{3+}$–$Nb^{5+}$, which might explain the limited first-cycle capacity of a previously reported Li-V-Nb-O DRX, which continuously increases upon cycling as the structure gets more disordered with V migration[39]. Overall, in all cases, compounds containing $Ti^{4+}$ and $Mo^{6+}$ lead to higher accessible Li contents than those containing $Nb^{5+}$. Moving to

divalent TM ions such as $Ni^{2+}$ and $Co^{2+}$, we find that they generally have higher accessible Li contents than the trivalent analogues, except that $Mn^{2+}$–$Nb^{5+}$ appears to be a poor-performing exception.

## Discussion

We have shown that SRO is critical in controlling Li-conductive environments and has a general dependence on chemistry. The remaining questions are thus what the microscopic origin of these trends is and how we can predict and manipulate SRO for the benefit of Li transport. Although the use of MC simulation is necessary to precisely reproduce SRO, empirical rules can be derived for intuitive prediction.

We find that the charge and size effects, which determine the stability of solid-state materials, also explain the trends in SRO. On the one hand, the high-valent TMs (e.g., $Mn^{3+}$, $Ti^{4+}$, $Nb^{5+}$) in DRXs tend to repel each other and intimately mix with $Li^+$ in order to keep local electroneutrality, thereby inhibiting Li segregation into $Li_4$ tetrahedra. This charge effect becomes more pronounced with increasing metal valence, based on nearest-neighbor pair analysis (Supplementary Figure 10 and Supplementary Method 3). On the other hand, the size mismatch between high-valent TMs and $Li^+$ facilitates Li segregation in order to minimize strain, counteracting the charge effect. The competition between the two effects is best demonstrated in the case of LMTO and LMZO, where $Zr^{4+}$ exhibits a stronger net attraction to $Li^+$ than $Ti^{4+}$ despite their common valence (Supplementary Figure 10). One explanation for this phenomenon is that $Ti^{4+}$ (0.605 Å) is much smaller than $Li^+$ (0.76 Å), and therefore, the size effect tends to segregate $Li^+$ from $Ti^{4+}$. In contrast, the size of $Zr^{4+}$ (0.72 Å) is close to that of $Li^+$, meaning that electrostatics dominates the size effect, favoring a maximal separation between the high-valent $Zr^{4+}$ and a corresponding local ordering between $Zr^{4+}$ and $Li^+$.

However, the prediction of SRO becomes elusive when comparing DRXs with non-isoelectronic TMs, e.g., $Li_{1.2}Mn_{0.4}Zr_{0.4}O_2$ and $Li_{1.2}Mn_{0.6}Nb_{0.2}O_2$. In this case, there is a more complex tradeoff between interaction strengths and TM concentration in

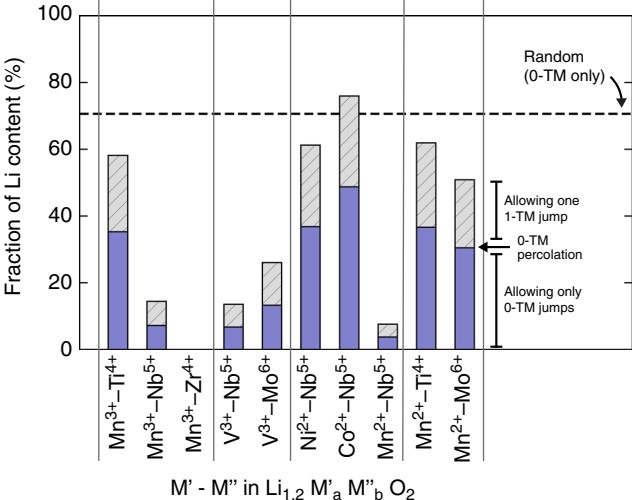

**Fig. 5** Fraction of Li content made accessible by the percolating network in a specific DRX $Li_{1.2}M'_aM''_bO_2$. Two conditions are considered: allowing only 0-TM jumps (blue), or allowing any given Li to make a single 1-TM jump before reaching the 0-TM percolating network (gray). A percolating Li level is considered the lower bound of accessible Li. The dotted line marks the fraction of Li accessible within the 0-TM percolating network in the random structure limit. The stoichiometry of each $Li_{1.2}M'_aM''_bO_2$ compound is constructed such that charge neutrality is retained. The various combinations of TM species M'–M'' are indicated along the x-axis

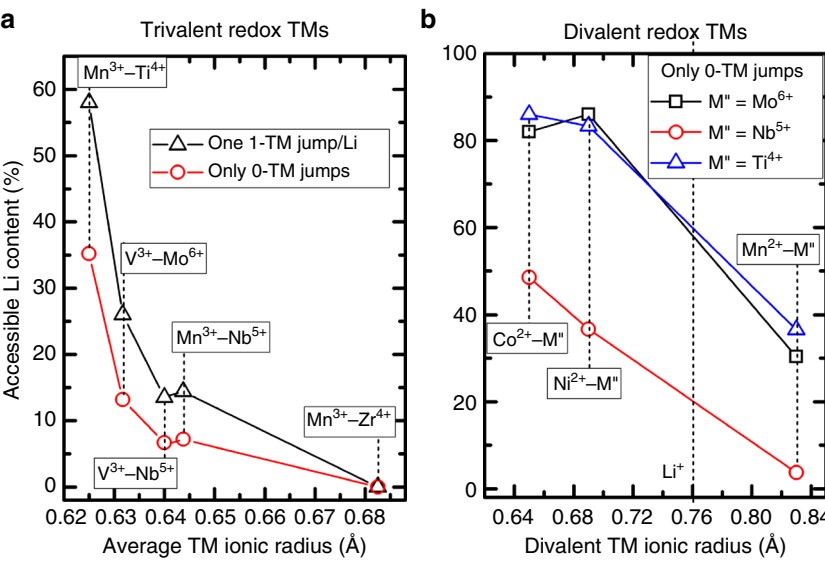

**Fig. 6** Correlation between accessible Li contents and ionic radii in various DRXs. **a** Accessible Li content as a function of the average TM ionic radius in DRXs composed of trivalent redox-active TMs, by allowing only 0-TM jumps (red) or allowing any given Li to make a single 1-TM jump before reaching the 0-TM percolating network (black). **b** Accessible Li content as a function of the divalent TM ionic radius in DRXs with divalent redox-active TMs and various stabilizers, by allowing only 0-TM jumps. The ionic radius of $Li^+$ is marked at 0.76 Å. All the compositions listed have the same Li-excess level of 20%

determining the degree of local ordering and Li segregation. Figure 6a shows a more intuitive relationship between accessible Li and chemistry, for DRXs composed of trivalent redox-active TMs. We observe a consistent negative correlation between the accessible Li content and the average TM ionic radius. The rationale behind the chosen $x$-axis is that in these DRXs, with a fixed Li-excess level, the average valence of the TMs is fixed accordingly and so is the electrostatic interaction strength between Li and TMs. Consequently, the size effect becomes dominant. Based on this correlation, we can further predict that high-valent metal species with large ionic sizes, e.g., $Sc^{3+}$ (0.745 Å) and $In^{3+}$ (0.80 Å), are likely to mix with $Li^+$ and impede Li diffusion, whereas others with small sizes, e.g., $Ga^{3+}$ (0.62 Å) and $Sb^{5+}$ (0.60 Å), are likely to facilitate local Li segregation and efficient transport.

DRXs containing divalent TMs form another unique class because divalent ions have the proper average metal valence in a DRX oxide and therefore do not require mixing with $Li^+$ to keep local electroneutrality. Additionally, they do not repel high-valent metals as much as trivalent ions do (Supplementary Figure 10), but can in turn mix with the high-valent metals to buffer their attraction to $Li^+$. This buffer effect explains why DRXs containing divalent TMs generally have better $Li_4$ segregation (Supplementary Figure 11) and higher accessible Li contents (Fig. 5) than their trivalent analogues. Figure 6b shows the accessible Li contents of DRXs containing divalent redox-active TMs and various stabilizers. We find that compounds containing $Ti^{4+}$ and $Mo^{6+}$ generally have higher accessible Li contents than the ones containing $Nb^{5+}$, a phenomenon possibly explained by the tradeoff between the electrostatic interaction strength and metal concentration. In addition, within each plot, we observe a consistent dependence on the divalent-TM radius. A feasible mechanism would be that as the ionic radius of the divalent TM increases, the increasing size mismatch tends to segregate the divalent TMs from the high-valent TMs, weakening the buffer effect, which eventually vanishes for $Mn^{2+}$, because its ionic radius is even larger than that of $Li^+$. From Fig. 6b, we also notice that some of the metal pairs with the highest predicted percolating Li levels, i.e. $Ni^{2+}$–$Ti^{4+}$ and $Ni^{2+}$–$Mo^{6+}$, have been experimentally realized in a compound $Li_{1.2}Ni_{1/3}Ti_{1/3}Mo_{2/15}O_2$ to deliver a high reversible capacity of 250 mAh g$^{-1}$ when cycled at 10 mA g$^{-1}$ between 1.5 and 4.5 V at room temperature[7]. However, it is worth noting that favorable cation SRO and a high kinetically accessible Li content only serve as necessary conditions and yet are insufficient to guarantee good capacity, which relies on various other factors, including the TM capacity, electronic conductivity, particle size, etc. For instance, although we predict a high kinetically accessible Li content >80% for $Li_{1.2}Ni_{0.2}Ti_{0.6}O_2$ (Fig. 6b), this compound is unlikely to deliver a high reversible capacity given its limited Ni content and the resulting small theoretical TM capacity.

Overall, we find that SRO controls the Li transport in DRXs by altering the distribution of 0-TM, 1-TM, and 2-TM channels as well as the connectivity between them. This observation is in contrast to stoichiometric layered oxides (e.g., $LiNi_{0.5}Mn_{0.5}O_2$[40] and $Li[Ni_xMn_xCo_{1-2x}]O_2$[41,42]) where Li and TMs are well separated and any SRO in the TM layer imposes minimal impact on overall Li transport kinetics. We have also identified a few guidelines for the manipulation of SRO. (i) For DRXs containing high-valent metals, the average TM ionic radius is an important metric to measure the degree of Li segregation. The use of large metal ions such as $Zr^{4+}$, $Sc^{3+}$, and $In^{3+}$ should be minimized. (ii) DRXs containing divalent TMs often facilitate Li segregation compared to their trivalent analogues due to the buffer effect of divalent TMs. This effect weakens as the ionic radius of the divalent TM increases. Therefore, small divalent TMs such as

$Co^{2+}$, $Ni^{2+}$ are more favorable than $Mn^{2+}$ and possibly $Fe^{2+}$, $Cu^{2+}$, $Zn^{2+}$. These guidelines are contrary to the common intuition that larger metal ions expand the lattice and are therefore favorable for Li transport. Furthermore, given that SRO differs significantly with compositions and fully controls the local environment around both Li and O ions, we expect that voltage profiles as well as the conditions for oxygen redox will also be modified by SRO. We thus propose that the manipulation of SRO can also be relevant for the design and engineering of voltage profiles and oxygen redox, thereby enabling future optimization of this class of new cathode materials with unprecedentedly high capacities and compositional flexibility. It should also be noted that TM rearrangement may occur upon prolonged cycling, which might modify the nature of SRO somewhat during subsequent cycles.

In conclusion, motivated by an experimental puzzle where two extremely similar compounds exhibit different electrochemical performance, we prove that cation short-range order (SRO) exists in long-range-disordered rocksalt cathodes. We have demonstrated how the SRO controls Li transport through altering local and macroscopic environments. More generally, we observe that electrostatics and ionic sizes strongly affect the SRO in cation-disordered rocksalts of different chemistries. Our findings uncover an important direction for future engineering and optimization of cation-disordered cathode materials to achieve higher capacities and energy densities.

## Methods

**Synthesis**. To synthesize LMTO and LMZO, stoichiometric $Mn_2O_3$, $TiO_2$, Zr $(OH)_4$, and $Li_2CO_3$ (with 5% excess) were dispersed into ethanol and thoroughly mixed using a planetary ball mill (Retsch PM 200) at 300 rpm for 16 h. The mixture was then dried, pelletized, and calcinated at 1100 °C in an argon atmosphere for 2 h, followed by furnace cooling.

**Characterization**. The XRD patterns were collected using a Bruker D8 ADVANCE diffractometer equipped with a Cu Kα radiation and an energy dispersive compound silicon strip LYNXEYE XE-T detector in the $2\theta$ range of 10–85°. The energy resolution of the detector completely filters out any Fe- or Mn-fluorescence, providing better resolution than a conventional secondary monochromator. The step width and collection time is 0.01° and 2.8 s, respectively. Rietveld refinement was performed using the HighScore Plus software package. Elemental analysis was performed by Luvak Inc. using direct-current plasma emission spectroscopy (ASTM E 1097-12) for the quantitative identification of metal species. The oxygen contents were confirmed using the inert gas fusion method (ASTM E 1019-11). SEM images were obtained on a Zeiss Gemini Ultra-55 analytical field-emission scanning electron microscope. ED patterns were taken after a grain was oriented properly on JEM-2100F using selected area electron diffraction in TEM mode. Time-of-flight (TOF) neutron powder diffraction was performed at room temperature on the Nanoscale Ordered Materials Diffractometer (NOMAD) at the Spallation Neutron Source at Oak Ridge National Laboratory. The samples for the neutron experiment were synthesized using a 7-Li enriched $Li_2CO_3$ source. The PDF patterns were analyzed using the PDFGui software package[43].

**Electrochemistry**. To fabricate electrodes, the product powder was first shaker-milled (SPEX 8000) for 1 h in an argon atmosphere. The milled active material (70 wt%) was then manually mixed with Super C65 carbon black (Timcal, 20 wt%) in a mortar for 30 min, followed by mixing with polytetrafluoroethylene (Dupont, 10 wt %) in an Ar-filled glovebox. The mixture was then rolled into a thin film to be used as the cathode. A coin cell was assembled using 1 M $LiPF_6$ (in a volumetric 1:1 mixture of ethylene carbonate and dimethyl carbonate), glass microfiber filters (grade GF/F, Whatman), and Li metal foil as the electrolyte, separator, and anode, respectively. The coin cells were tested on an Arbin battery testing station. PITT measurements were performed on a Solartron Analytical 1470E Celltest System.

**DFT calculations**. All the DFT calculations were performed using the Vienna ab Initio simulation package (VASP)[44,45] with projector-augmented wave[46] pseudo-potentials and the exchange-correlation functional by Perdew, Burke, and Ernzerhof[47]. To correct the DFT self-interaction error, the Hubbard-U correction[48] was employed for the transition-metal $d$ states where needed with values taken from Jain et al.[49]. The U parameters were originally fitted to oxidation enthalpies that have been shown to ensure the robust prediction of relative energies[50]. We employed $k$-point meshes with a reciprocal spacing of 25 $k$-points per Å$^{-1}$ for the

Brillouin-zone integration and a plane wave basis set with an energy cutoff of 520 eV. All the DFT energies and atomic forces were converged to 0.001 meV per atom and 20 meV Å$^{-1}$, respectively. Input files for the DFT calculations were generated using the Python Materials Genomics[51] package.

**Cluster expansion**. The cluster expansion (CE) method is used in the present work to enable extensive configurational sampling that would not directly be possible with DFT. The CE method is a lattice model that captures the configurational degrees of freedom (which are the relevant quantity in the present work). CE is based on a formally exact expansion of the formation energy in contributions from different types of interaction between lattice sites. Our CE model was found to reproduce DFT energies with good accuracy (<8 meV per atom) when pair interactions (two lattice sites), triplet interactions (three sites), and quadruplets (four sites) up to the cutoff distances are included in the expansion. The CE model was constructed by fitting to around 500 lattice configurations and their DFT energies. The accuracy was determined by cross-validation, i.e., based on atomic configurations that were not included in the fit. More details are given below.

Cluster expansion Hamiltonians[52] for each chemical space discussed here were constructed based on the energies of ~500 lattice configurations, where the energies were computed with DFT. Each cluster expansion relied on a basis set of pair interactions up to 7 Å, triplet interactions up to 4.1 Å, and quadruplet interactions up to 4.1 Å, with respect to a rocksalt primitive cell with lattice constant $a = 3$ Å, on top of a baseline of formal-charge electrostatics and a fitted dielectric constant[33]. The effective cluster interactions and dielectric constant were obtained from a L$_1$-regularized linear regression fit, with the regularization parameter optimized by cross-validation[53]. The resulting fits yielded an out-of-sample root mean square error of less than 8 meV per atom. All canonical Monte Carlo simulations based on these Hamiltonians were run using the Metropolis-Hastings algorithm.

While cluster expansion models can in principle include the effect of phonon entropy[54,55], this is rarely done in practice when predicting finite-temperature phase stability as the variation of vibrational entropy with short and long-range order of the cations is much smaller than the effect of configurational entropy[56]. Thus, the effect of lattice dynamics on the cation configuration, and thereby the population of 0-TM, 1-TM, and other Li-TM clusters, can be expected to be negligible compared to the effect of configurational free energy which we sample accurately.

For the connectivity analysis, the definitions of 0-TM, 1-TM, and 2-TM channels of reference[4] were used. For each composition and temperature, the connectivity was averaged over 600 atomic configurations obtained from MC simulations as previously described. To ensure convergence of the 0-TM Li network size, $3 \times 3 \times 3$ supercells of the MC configurations containing 12,960 cation sites were used.

**Simulation of ED patterns**. The LMZO and LMTO atomic configurations equilibrated with Monte Carlo simulations at 1000 °C as described above were used to simulate electron diffraction (ED) images. For this purpose, cubic sections with an edge length of 50 Å (>12,000 atoms) were truncated from the periodic bulk structures defined by 960-atom unit cells. Electron diffraction patterns for the cubic [100] zone axis were computed for each cubic cell using the methods and potentials given by Kirkland[57], using the potential calculation method described in reference[58]. The final diffraction images were calculated as an incoherent sum of all 500 simulated patterns, and these summed images were smoothed and scaled by amplitude (square root of intensity) to more clearly show the SRO features of the pattern.

## Data availability
The datasets generated and analyzed during the current study are available from the corresponding author on reasonable request.

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

## Acknowledgements

This work was supported by the Robert Bosch Corporation, Umicore Specialty Oxides and Chemicals, and the Assistant Secretary for Energy Efficiency and Renewable Energy, Vehicle Technologies Office of the U.S. Department of Energy under Contract No. DE-AC02-05CH11231 under the Advanced Battery Materials Research (BMR) Program. The research conducted at the NOMAD Beamline at ORNL's Spallation Neutron Source was sponsored by the Scientific User Facilities Division, Office of Basic Sciences, U.S. Department of Energy. Work at the Molecular Foundry was supported by the Office of Science, Office of Basic Energy Sciences, of the U.S. Department of Energy under Contract No. DE-AC02-05CH11231. The computational analysis was performed using computational resources sponsored by the Department of Energy's Office of Energy Efficiency and Renewable Energy and located at the National Renewable Energy Laboratory, as well computational resources provided by Extreme Science and Engineering Discovery Environment (XSEDE), which was supported by the National Science Foundation grant number ACI-1053575. The authors would like to thank Penghao Xiao, Jue Liu, Jinhyuk Lee, and Yuanpeng Zhang for helpful discussion.

## Author contributions

G.C. and H.J. designed and planned the study. G.C. supervised all aspects of the research. H.J. synthesized, characterized, and electrochemically tested the compounds. A.U. performed percolation and connectivity analysis. D.A.K. performed DFT and cluster expansion calculations. D.-H.K. acquired ED data. N.A., C.O., and D.-H.K. simulated diffuse scattering patterns. W.H. computed voltage profiles. Z.C. performed supportive synthesis and electrochemical experiments. T.S. performed SEM. H.J. and J.C.K. performed PITT tests. The manuscript was written by H.J. and G.C. and was revised by A.U. and D.A.K. with the help of J.C.K. and H.K. All authors contributed to discussions and commented on the manuscript.

## Additional information

**Competing interest** The authors declare no competing interests.

