## [Peer Review File · Nature Communications]

Reviewers' comments:

Reviewer #1 (Remarks to the Author):

Nature Communications manuscript NCOMMS-18-26372-T

The paper by Ji et al. deals with the field of energy storage and more specifically targets the complex issue of Li-ion transport in cation-disordered oxides for rechargeable Li-ion batteries. This topic is relevant to the field of battery but it is extremely complex. It is thus great to see theorists willing to address this issue. The thoughts brought into this manuscript are interesting; therefore the paper raised several questions and concerns that must be thoroughly addressed prior it can be considered for publication.

My main concerns are:

- 1) Figure 2: (What is the cycling rate ?). Not only the capacity but also the voltage amplitude and profile are significantly modified in LMZO between the room- and high-temperature measurements. Unless the authors show that equivalent thermodynamic pathways / mechanisms are achieved at both temperatures, it cannot be stated that "the capacity of LMZO is limited by Li transport kinetics". In addition, if kinetic limitations exist, I would expect polarization effects to lead to higher potential (in charge) and lower potential (in discharge) for the RT measurements. Though, the potential in discharge is significantly lower at 50° than at RT. Can the authors discuss this point ?
- 2) In line with question 1) we advise the authors to normalize their RT and 50°C measurements to better comment their measurements. That is replotting the data with the capacity at 50°C matching that of the RT. Voltage hysteresis differences, if any, will then clearly appear.
- 3) The Monte Carlo expansion cluster calculations is well done and they clearly demonstrate that the titanium compound has a better network (no local order) for the diffusion of lithium than the zirconium compound which has a local cationic order more pronounced which limits the percolation of Li. However, the measurements of diffusion coefficients GITT are MACROSCOPIC so cannot be compared for example with the kinetic activation barriers that are calculated in DFT (NEB calculations) on very short range paths. To circumvent this, authors have used Monte Carlo model. Whatever, the main issue with GITT is that the method is not valid for two phases system but more so that it take it take into account diffusion at the grain boundaries which can be sometimes more limiting than those in the bulk **but which are not taken into account in the calculations**. As the morphology of the two electrodes LMZO and LMTO is quite similar, the authors have probably thought that the diffusion at the grain boundaries is equivalent in the two compounds. First, this is not at all discussed in the manuscript. Secondly, the authors should proof that grain boundaries are negligible. Additional measurements such as AC-impedance spectroscopy or NMR are recommended.
- 4) More information about the theoretical methodology are required to allow the large audience of nature communications to understand the results of calculations. As far as I understood, the cluster expansion model is built on DFT calculations performed on the lithiated phases. From the results it is clear that the lithiated LMTO has better Li-diffusivity than LMZO. **But how this evolves upon delithiation, in particular in systems that are known to undergo significant structural reorganizations upon cycling?** The

difference in the diffusion coefficients in the two compounds is much more pronounced upon delithiation than in the pristine materials.

- 5) page 7 : Zr⁴⁺ and Ti⁴⁺ are said to have "common valence" and that mainly the size of the cation governs SRO. While the two cations have the same "formal" oxidation state, **their valence orbitals are completely different which clearly affect the electron density around Ti⁴⁺ (involved in Ti(d)/O(p) covalent interactions) compared to Zr⁴⁺ (mainly electrostatic interaction)**. Neglecting chemical bonding and reasoning only on electrostatics might be dangerous. Ru⁴⁺ for instance has an ionic radius similar to Li⁺ and is known in the Li₂RuO₃ phase to form a layered rather than a DRX phase. So in light of this comment, I cannot buy into the enumerated guidelines simply based on charge and size consideration. Please comment on this point.
- 6) DRX compounds offer great interest for fundamental studies but let's recall that in terms of kinetics they behave quite poorly than classical and well-ordered Li-rich NMC materials. It is the reason why cycling data for DRX materials is frequently collected at 50°C (see Yabuchi). Could the authors discuss this point?

Reviewer #2 (Remarks to the Author):

This work presents an interesting study of local short range of structural ordering in cation-disordered oxide cathodes and its impact on electrochemical performances for lithium-ion batteries. It is known that the local structural ordering plays an important role in determining materials properties, brings in diverse lithium storage behavior for oxide cathodes but adds complexities for structural studies. The authors combined multiple techniques including electron diffraction, neutron pair distribution function and Monte Carlo simulation to study the short range of ordering (SRO) in $\text{Li}_{1.2}\text{Mn}_{0.4}\text{Ti}_{0.4}\text{O}_2$ (LMTO) and $\text{Li}_{1.2}\text{Mn}_{0.4}\text{Zr}_{0.4}\text{O}_2$, and claimed that the different SRO cause different lithium diffusion behavior and thus the distinct electrochemical performances. However, this manuscript didn't provide a clear picture for the proposed SRO. Therefore, revisions may need before this manuscript could be further considered for publication in Nature Communications. I have several comments for the authors to consider:

(1) The authors claimed the different short range of ordering (SRO) between LMTO and LMZO, however, the authors didn't provide a clear picture for these SRO. What is the difference of SRO between LMTO and LMZO? Is it deduced from the ED patterns? If so, the ED is highly selective and may not provide sufficient representative. This is the core of this manuscript and definitely needs further interpretations.

(2) There is no doubt that the local ordering will affect the macroscopic lithium transport behavior and thus the electrochemical performances. However, it is a bit arbitrary to solely attribute the different "observed capacity" between LiMnTiO_2 and LiMnZrO_2 to the different SRO, as the electrochemical behavior is normally affected by many aspects. For example, (a) the real chemical composition of the materials deviates a little bit from the designed composition, will it cause the differences in observed capacity as the lithium content has significant impact on lithium storage capacity for cation-disordered cathode, based on the authors' percolation theory; (b) Is there any experiment evidence to show that both LMTO and LMZO have the similar surface structure? Any surface impurity phases? A notable bump at around 22 degree (Figure 1c) and additional peak at 28 degree (Figure S1) can be seen on XRD pattern for LMZO, quite different from LMTO.

In addition, the major difference of the charge-discharge voltage profiles (Figure 2a and 2b) appears in the high voltage region (above 4V, $x > 0.5$), and this cannot be simply explained by the different Li^+ diffusivity induced by the different SRO.

(3) Following by the previous question, the high voltage plateau for LMTO is quite similar to that of layered Li rich oxide with Li_2MnO_3 component, is there any possibility for the Li_2MnO_3 -typed SRO? If not, what the origin for this high voltage plateau?

(4) Based on the presented results, the SRO will reduced the content of Li_4 clusters that in favor of lithium diffusion. Is there possibility that this negative impact be counteracted by increasing the lithium content in pristine materials? Any experiment results to support?

(5) The authors proposed some general rules for selecting high performance cation-disordered oxide with desired SRO as displayed in Figure 5, then the reviewer wishes to see the actual experimental

data with the calculated most improved material.

(6) Please note that the term “diffusion coefficient” labelled on Figure 2d should be “apparent diffusion coefficient”, which cannot be directly linked to the lithium diffusion between atomic sites.

Response to the Reviewers: Manuscript ID NCOMMS-18-26372-T

We thank the editor and both reviewers for their time spent to evaluate our manuscript and for their comments. All of the reviewers' concerns are addressed below in detail. All changes made in the manuscript are **highlighted in blue**. The revised manuscript has also included additional figure **XXX** as discussed below.

Response to Reviewer #1

Nature Communications manuscript NCOMMS-18-26372-T The paper by Ji et al. deals with the field of energy storage and more specifically targets the complex issue of Li-ion transport in cation-disordered oxides for rechargeable Li-ion batteries. This topic is relevant to the field of battery but it is extremely complex. It is thus great to see theorists willing to address this issue. The thoughts brought into this manuscript are interesting; therefore the paper raised several questions and concerns that must be thoroughly addressed prior it can be considered for publication.

We thank Reviewer #1 for the positive evaluation of our manuscript and thoughtful questions. All concerns are addressed in detail below.

The first three questions of Reviewer #1 are about the validity of our experimental methods. While we individually addressed each question, we would like to point out that the main purpose of the two experiments in question (high-temperature cycling and PITT) is to demonstrate that LMTO and LMZO have very different Li transport kinetics despite their extreme similarity in structure and chemistry, and that this kinetic difference is what guided us to discover the different SRO types as well as their very different amounts of kinetically extractable Li. It is never possible in battery research to characterize every possible aspect of a material (surface structure, detailed bulk structure, etc.) but we argue that all theoretical and experimental evidence we acquired is consistent with the kinetics of these two materials being different. In the revised manuscript, we furthermore add new experimental data that compares the rate capability of both materials, as shown in the answer to Question #3, to further demonstrate the kinetic differences between LMTO and LMZO.

My main concerns are:

1) Figure 2: (What is the cycling rate?). Not only the capacity but also the voltage amplitude and profile are significantly modified in LMZO between the room- and high temperature measurements. Unless the authors show that equivalent thermodynamic pathways / mechanisms are achieved at both temperatures, it cannot be stated that "the capacity of LMZO is limited by Li transport kinetics". In addition, if kinetic limitations exist, I would expect polarization effects to lead to higher potential (in charge) and lower potential (in discharge) for the RT measurements. Though, the potential in discharge is significantly lower at 50° than at RT. Can the authors discuss this point?

Answer: The cycling rate shown in Figure 2 for both temperatures is 10 mA/g and is updated in the caption of Figure 2 in manuscript.

We appreciate the reviewer’s attention to the potential change in going from RT to 50 °C. We find that the difference between the voltage profiles of LMZO at room temperature and high temperature is due to reaction that takes place during charge between 4.5 and 4.7 V at 50 °C. In a replot of the LMZO data from Figure 2 (shown in Figure R1 below), an extra plateau-like feature is observed above 4.5 V at 50 °C, which may result from reaction of the cathode with the electrolyte due to the highly oxidizing condition. As a result, the subsequent discharge profile becomes steep with larger polarization. This extra plateau feature is less pronounced in the RT profile. We tested multiple cells between 1.5 – 4.7 V at 50 °C and confirmed that they all behave similarly.

To avoid the complexity induced by the high-voltage, high-temperature side reaction, we cycled LMZO between 1.5 – 4.5 V at 50 °C, and the data is shown in Figure R2 below. For comparison, cycling in the same voltage range at RT is also shown. The extra plateau-like feature does not show up at high temperature and the subsequent discharge profile clearly has the same shape as the RT one, as argued by the reviewer. Also, as expected for a kinetically limited cathode material, the high-temperature discharge profile has a significantly higher voltage than the RT one.

We thus chose to replot Figure 2a and 2b with the discharge profile starting from the same point at 0 to allow for a better comparison of the polarization, and with updated data for LMZO between 1.5–4.5 V. We added the following sentence to the description of the experiment in manuscript: “The upper cutoff voltage for LMZO is set at 4.5 V, while that of LMTO is set at 4.7 V. This is because we find that cycling LMZO to 4.7 V at 50 °C provides little extra reversible capacity compared to a 4.5 V-cutoff and causes a side-reaction above 4.5 V which seems to lead to increased polarization and subsequent discharge voltage fade.” We also updated the capacities of LMZO between 1.5–4.5 V at RT (0.57 Li/f.u.) and at 50 °C (0.87 Li/f.u.) in the new Figure 2e as well as in manuscript.

Figure R1: (Replotted from data in the original Figure 2a) First-cycle galvanostatic voltage profiles of LMZO at room temperature (black) and 50 °C (red) between 1.5–4.7 V at 10 mA/g.

Figure R2: (Newly collected data) First-cycle galvanostatic voltage profiles of LMZO at room temperature (black) and 50 °C (red) between 1.5–4.5 V at 10 mA/g.

2) *In line with question 1) we advise the authors to normalize their RT and 50°C measurements to better comment their measurements. That is replotting the data with the capacity at 50°C matching that of the RT. Voltage hysteresis differences, if any, will then clearly appear.*

Answer: As suggested by the reviewer, we replot the data of LMZO and LMTO at both temperatures by normalizing the charging capacity at 50 °C to that at room temperature. For the voltage profiles of LMZO, we use the newly collected data between 1.5–4.5 V. Although capacity normalization is more useful to visualize polarization for materials with plateaus but less useful for DRXs with very steep voltage profiles, we still observe more significant polarization between RT and high-temperature voltage profiles, especially during discharge where any kinetic limitation is most pronounced, for LMZO than for LMTO.

Figure R3: A replot of the original Figure 2a and 2b by normalizing the charging capacity at 50°C to that at room temperature. The charging capacity at room temperature is set to 1. The data of LMZO is newly collected between 1.5–4.5 V.

3) *The Monte Carlo expansion cluster calculations is well done and they clearly demonstrate that the titanium compound has a better network (no local order) for the diffusion of lithium than the zirconium compound which has a local cationic order more pronounced which limits the percolation of Li. However, the measurements of diffusion coefficients GITT are MACROSCOPIC so cannot be compared for example with the kinetic activation barriers that are calculated in DFT (NEB calculations) on very short range paths. To circumvent this, authors have used Monte Carlo model. Whatever, the main issue with GITT is that the method is not valid for two phases system but more so that it take it take into account diffusion at the grain boundaries which can be sometimes more limiting than those in the bulk but which are not taken into account in the calculations. As the morphology of the two electrodes LMZO and LMTO is quite similar, the authors have probably thought that the diffusion at the grain boundaries is equivalent in the two compounds. First, this is not at all discussed in the manuscript. Secondly, the authors should proof that grain boundaries are negligible. Additional measurements such as AC impedance spectroscopy or NMR are recommended.*

Answer: We do disagree with the reviewer on some of the points he/she makes. Macroscopic diffusion can be related to the local migration energy of ions. Macroscopic diffusion arises by a series of local hops. As such, the migration barrier for an ion is the key quantity in setting the diffusion constant. This is reflected in the theory of diffusion, which relates the macroscopic diffusivity to various factors, including the exponential of

the migration energy. But we do agree on the points made regarding possible effects of grain boundary diffusion and the issue of PITT in two-phase regions, The purpose of the PITT experiment is to further corroborate the high-temperature *vs.* room-temperature cycling experiment to prove that LMZO is more kinetically limited than LMTO, and both data sets do come to the same conclusion.

We agree with the reviewer that the PITT-deduced Li diffusivity is not accurate for two-phase systems but more reliable for solid solutions. Because the phase change behavior for $x > 4$ is unclear we decided to remove the part in our Li diffusivity data after $x = 0.4$, which is the theoretical Mn capacity in LMTO and LMZO. This is motivated by previous studies on a similar DRX material, $\text{Li}_{1.3}\text{Mn}_{0.4}\text{Nb}_{0.4}\text{O}_2$, that show that $\text{Mn}^{3+}/\text{Mn}^{4+}$ oxidation happens first during initial charge and is followed by oxygen oxidation and that the $\text{Mn}^{3+}/\text{Mn}^{4+}$ oxidation region correlates with a solid solution reaction while the oxygen oxidation region seems to be more complicated and may have two-phase-like behavior [Kan, Wang Hay, et al. *Chemistry of Materials* 30.5 (2018): 1655-1666.].

Regarding the Li diffusion at grain boundaries, this is certainly an important question but one that is not easy to address quantitatively, either with modeling or experiments. In most battery materials bulk diffusion is fast enough so that parallel diffusion along grain boundaries does not significantly add to the total diffusivity. Cross-boundary diffusion could in principle be limiting, but there is no indication in our data that it is. If boundary diffusion were limiting one would expect a fairly constant “resistance” against intercalation, and not the variable diffusion constant with composition that is observed. We stress that the PITT experiment is one of multiple pieces of data, computed or experimental, which all point towards poorer bulk kinetics in the LMZO material.

We also added the following discussion to the description of the PITT data: “It should be noted that the apparent Li diffusivity deduced from PITT include the contribution from bulk and grain boundaries whereas the migration barriers from nudged elastic band calculations would only impact the bulk diffusivity. We observe that the Li diffusivity in LMTO is much higher than that in LMZO, confirming that the capacity of LMZO is more limited by Li transport kinetics. Given the validity of the PITT method in solid solution systems, we plot the region from $x = 0$ to $x = 0.4$ where $\text{Mn}^{3+/4+}$ oxidation and a solid-solution reaction dominate [Kan, Wang Hay, et al. *Chemistry of Materials* 30.5 (2018): 1655-1666; Yabuuchi, Naoaki, et al. *Nature communications* 7 (2016): 13814.]”

To further substantiate this interpretation of our data, we have performed an additional rate experiment the results of which are shown in the two figures below (added as new Figure 2a and 2b). The rate capability of LMTO is clearly better than the one of LMZO. This observation further supports our conclusion that LMZO is significantly more kinetically limited than LMTO, although based on conventional arguments one would expect LMZO to exhibit better kinetics given its larger lattice parameters. We also added a sentence to the corresponding results section: “Indeed, the rate capability of LMTO (Figure 2b) is clearly better than that of LMZO (Figure 2a).”

Figure caption: First-cycle voltage profiles of LMZO (left) and LMTO (right) when cycled between 1.5–4.7 V at 10, 40, 130 and 260 mA/g.

4) More information about the theoretical methodology are required to allow the large audience of nature communications to understand the results of calculations. As far as I understood, the cluster expansion model is built on DFT calculations performed on the lithiated phases. From the results it is clear that the lithiated LMTO has better Li diffusivity than LMZO. But how this evolves upon delithiation, in particular in systems that are known to undergo significant structural reorganizations upon cycling? The difference in the diffusion coefficients in the two compounds is much more pronounced upon delithiation than in the pristine materials.

Answer: The cluster expansion (CE) method is used in the present work to enable extensive configurational sampling that would not directly be possible with DFT. As further detailed in the methods section, the CE method is a lattice model that captures the configurational degrees of freedom (which are the relevant quantity in the present work). CE is based on a formally exact expansion of the formation energy in contributions from different types of interaction between lattice sites. Our CE model was found to reproduce DFT energies with good accuracy (< 8 meV/atom) when pair interactions (two lattice sites), triplet interactions (three sites), and quadruplets (four sites) up to the cutoff distances given in the methods section are included in the expansion. As detailed in the methods section, the CE model was constructed by fitting to around 500 lattice configurations and their DFT energies. The accuracy was determined by cross validation, i.e., based on atomic configurations that were not included in the fit.

And the following paragraph is added to the Methods section about “Cluster expansion”:
 “The cluster expansion (CE) method is used in the present work to enable extensive configurational sampling that would not directly be possible with DFT. The CE method is a lattice model that captures the configurational degrees of freedom (which are the relevant quantity in the present work). CE is based on a formally exact expansion of the formation energy in contributions from different types of interaction between lattice sites. Our CE model was found to reproduce DFT energies with good accuracy (< 8 meV/atom) when pair interactions (two lattice sites), triplet interactions (three sites), and quadruplets (four sites) up to the cutoff distances are included in the expansion. The CE model was constructed by fitting to around 500 lattice configurations and their DFT energies. The accuracy was determined by cross validation, i.e., based on atomic configurations that were not included in the fit. More details are given below.”

To first order the Li diffusivity upon delithiation will behave qualitatively the same as when fully lithiated though the absolute value will be different. This is because the diffusivity is controlled by “0-TM” channels. In the lithiated state the sites around the 0-TM channel are occupied by Li, whereas upon delithiation they become more and more occupied by vacancies. However, they remain “0-TM” channels and therefore the percolation properties of the material are unchanged. This does not mean that the diffusion constant itself does not change as the actual migration rates depend on lattice parameter, frequency of vacancies etc. But the variation of percolation and diffusivity with short range order will remain largely unchanged.

This could change if transition metal migration were to occur upon cycling. But in general transition metal migration needs to overcome significantly higher energy barriers (> 1 eV [Reed and Ceder. "Role of electronic structure in the susceptibility of metastable transition-metal oxide structures to transformation." *Chemical reviews* 104.10 (2004): 4513-4534.]) compared to Li ions (< 300 meV [Lee et al. "Unlocking the potential of cation-disordered oxides for rechargeable lithium batteries." *Science* 343.6170 (2014): 519-522.]). However, we agree that short-range metal migration can occur when lithium is removed. This aspect of potential metal migration is not unique to cation-disordered cathodes and quite common in layered cathodes at high state of charge. This is an important research question to be investigated, but does not distract from the importance of cation short-range order which is studied in this manuscript.

We have commented on the possibility of transition metal migration and its potential impact on short-range order and Li transport to the end of the discussion section: “It should also be noted that TM rearrangement may occur upon prolonged cycling, which might modify the nature of SRO somewhat during subsequent cycles.”

Finally, the similar Li diffusivity of the two materials at the beginning of charge ($x < 0.05$) in Figure 2d is due to the large thermodynamic factor in the voltage profiles for both materials which swamps the self-diffusion. The chemical diffusivity, which is what is measured by PITT, is the product of the thermodynamic factor and the self-diffusivity. The latter is more representative of the intrinsic mobility of ions, whereas the thermodynamic factor includes the driving force provided by a concentration gradient.

5) page 7 : Zr⁴⁺ and Ti⁴⁺ are said to have "common valence" and that mainly the size of the cation governs SRO. While the two cations have the same "formal" oxidation state, their valence orbitals are completely different which clearly affect the electron density around Ti⁴⁺ (involved in Ti(d)/O(p) covalent interactions) compared to Zr⁴⁺ (mainly electrostatic interaction). Neglecting chemical bonding and reasoning only on electrostatics might be dangerous. Ru⁴⁺ for instance has an ionic radius similar to Li⁺ and is known in the Li₂RuO₃ phase to form a layered rather than a DRX phase. So in light of this comment, I cannot buy into the enumerated guidelines simply based on charge and size consideration. Please comment on this point.

Answer: We respectfully disagree with this comment. Our group has an extensive track record in understanding long range and short-range order in materials, based on whatever

are the relevant chemistry descriptors for a given situation. The statement that in the case of Zr and Ti, the size is the main differentiator in setting the SRO is a correct statement.

While the reviewer is right that the Ti 3*d* and Zr 4*d* bands are at different energies relative to the oxygen 2*p* band, this is irrelevant for Zr⁴⁺ and Ti⁴⁺ ions which are both *d*⁰ ions and have no electrons available for hybridization with the oxygen states. In fact we have recently shown that this is precisely the reason why *d*⁰ species promote cation disorder in Li transition-metal oxides (Urban et al., *Phys. Rev. Lett.* **119**, 2017, 176402). The Ru⁴⁺ example that the reviewer brings up is a *d*⁴ ion, which behaves chemically completely different and is unlikely to form a DRX by itself without a *d*⁰ ion.

We now added a sentence to the first paragraph in the section “Chemistry dependence of SRO and Li transport environments”: “Besides, since *d*⁰ TMs are known to promote cation disorder in Li-TM oxides [Urban et al., *Phys. Rev. Lett.* **119**, 2017, 176402], all the hypothetical compositions contain a redox-active TM as well as a *d*⁰ TM.”

6) DRX compounds offer great interest for fundamental studies but let's recall that in terms of kinetics they behave quite poorly that classical and well-ordered Li-rich NMC materials. It is the reason why cycling data for DRX materials is frequently collected at 50°C (see Yabuchi). Could the authors discuss this point?

Answer: We thank the reviewer for the critical comment. However, please note that we have reported several new DRX materials with reasonable rate capability and capacity at room temperature, e.g. Li_{1.2}Ni_{1/3}Ti_{1/3}Mo_{2/15}O₂, Li_{1.15}Ni_{0.45}Ti_{0.3}Mo_{0.1}O_{1.85}F_{0.15}, Li₂Mn_{1/2}Ti_{1/2}O₂F [Lee, Jinhyuk, et al. *Energy & Environmental Science* 8.11 (2015): 3255-3265; Lee, Jinhyuk, et al. *Nature communications* 8.1 (2017): 981; Lee, Jinhyuk, et al. *Nature* 556.7700 (2018): 185.]. Note that our data is almost always taken at room temperature, unlike the situation of Yabuchi's work which is brought up by the reviewer. The only 50°C data shown is for the purpose of better understanding the kinetic origin of the capacity limitation of LMZO. We agree that the rate capability of DRX materials is not quite yet at the level needed for commercialization, but the field of battery materials is rife with examples of materials that displayed poor rate capability at conception and which have been improved over the years, a good example, being LiFePO₄. This is exactly why a thorough understanding of any limiting factor to the rate capability is important before rational optimization can be achieved. This current work shows that unlike layered oxides, where Li transport largely happens in the Li layer and any transition metal ordering in a separate layer does not have a significant impact, cation short-range order is crucial in determining kinetically extractable Li in DRX materials as Li transport relies on percolating through a 3D network.

Response to Reviewer #2

This work presents an interesting study of local short range of structural ordering in cation-disordered oxide cathodes and its impact on electrochemical performances for lithium-ion batteries. It is known that the local structural ordering plays an important role in determining materials properties, brings in diverse lithium storage behavior for oxide cathodes but adds complexities for structural studies. The authors combined multiple techniques including electron diffraction, neutron pair distribution function and Monte Carlo simulation to study the short range of ordering (SRO) in $\text{Li}_{1.2}\text{Mn}_{0.4}\text{Ti}_{0.4}\text{O}_2$ (LMTO) and $\text{Li}_{1.2}\text{Mn}_{0.4}\text{Zr}_{0.4}\text{O}_2$, and claimed that the different SRO cause different lithium diffusion behavior and thus the distinct electrochemical performances. However, this manuscript didn't provide a clear picture for the proposed SRO. Therefore, revisions may need before this manuscript could be further considered for publication in Nature Communications. I have several comments for the authors to consider:

We appreciate Reviewer #2 for the positive evaluation and thoughtful questions.

(1) The authors claimed the different short range of ordering (SRO) between LMTO and LMZO, however, the authors didn't provide a clear picture for these SRO. What is the difference of SRO between LMTO and LMZO? Is it deduced from the ED patterns? If so, the ED is highly selective and may not provide sufficient representative. This is the core of this manuscript and definitely needs further interpretations.

Answer: The SRO is deduced from the combination of ED patterns, ab-initio Monte Carlo simulation and Neutron Pair Distribution Functions, all which show a consistent picture.

The nature of the SRO in both materials has been discussed in a substantial part of our manuscript, in the section “Computational modeling of Li transport environments in LMTO and LMZO”. In essence, locally, the SRO in LMTO favors the formation of Li_4 local clusters more than in LMZO, while LMZO favors the formation of Li_3M clusters (as shown in Figure 4a); macroscopically, the Li_4 clusters in LMTO are connected extensively while those in LMZO are scattered and poorly connected.

In order to provide a clearer picture of the SRO, we added the following paragraph to the end of the Section “Computational modeling of Li transport environments in LMTO and LMZO”: “We may draw some useful insight about the impact of SRO on the preferred cation configurations around tetrahedral sites based on the relationship between SRO and its corresponding ground states. This is because SRO is generally considered an “intermediate state” between a random state and an ordered low-temperature ground state, and therefore, it often resembles the low-temperature long-range order [De Ridder, R., Van Tendeloo, G. & Amelinckx, S. *Acta Crystallogr. Sect. A Cryst.* **32**, 216-224 (1976).]. In DRX oxides with multiple TM species, our calculations indicate that phase separation into ground-state compounds is preferred at low temperatures. For example, LMZO is thermodynamically favored to decompose into LiMnO_2 and Li_2ZrO_3 at low temperatures according to our calculations. Li_2ZrO_3 is known to have a $\gamma\text{-LiFeO}_2$ -like structure [Hodeau, J.L., Marezio, M., Santoro, A. & Roth, R.S. *J. Solid State Chem.*, **45**(2), 170-179 (1982).] in which most cation tetrahedra are made up out of 2Li and 2Fe ions, which gives the compound very poor Li percolation [Urban, A., Lee, J. & Ceder, G. *Adv. Energy Mater.* **4**, 1400478 (2014).]. Indeed, not even one Li site in Li_2ZrO_3 is

connected to other Li sites through 0-TM channels even at a Li-excess level of 33.3%. In contrast, Li_2TiO_3 adopts a Li_2MnO_3 -like structure [Dorrian, J.F. & Newnham, R.E. Mater. Res. Bull., 4(3), 179-183 (1969).] which contains a large amount of tetrahedra with 1TM and 3Li as well as tetrahedra with 4Li, which makes every Li site connected with other Li sites. The SRO and percolation properties observed in the cation-disordered materials are reminiscent of the low-temperature ground states, which explain why LMTO favors the segregation of Li in tetrahedra while LMZO exhibits poor Li conductivity.”

(2) There is no doubt that the local ordering will affect the macroscopic lithium transport behavior and thus the electrochemical performances. However, it is a bit arbitrary to solely attribute the different “observed capacity” between LiMnTiO_2 and LiMnZrO_2 to the different SRO, as the electrochemical behavior is normally affected by many aspects. For example, (a) the real chemical composition of the materials deviates a little bit from the designed composition, will it cause the differences in observed capacity as the lithium content has significant impact on lithium storage capacity for cation-disordered cathode, based on the authors’ percolation theory; (b) Is there any experiment evidence to show that both LMTO and LMZO have the similar surface structure? Any surface impurity phases? A notable bump at around 22 degree (Figure 1c) and additional peak at 28 degree (Figure S1) can be seen on XRD pattern for LMZO, quite different from LMTO.

In addition, the major difference of the charge-discharge voltage profiles (Figure 2a and 2b) appears in the high voltage region (above 4V, $x > 0.5$), and this cannot be simply explained by the different Li^+ diffusivity induced by the different SRO.

Answer: We agree that SRO-controlled Li transport alone may not account for all the difference between the two materials, but it is nevertheless a major factor at play. The clear difference in the Li transport kinetics between the two materials despite their high similarity in chemistry and crystal structure is what guides us to discover the critical role of SRO in controlling the amount of kinetically accessible Li. Besides, the computed voltage profiles of LMTO and LMZO achieved by enumerating a large number of possible configurations (Figure S4) indicate that both materials should cycle in a comparable voltage window and their capacity would not be limited by thermodynamics. We added the following sentence to the section of synthesis and electrochemistry: “especially in LMTO and LMZO. The computed voltage profiles obtained by enumerating a large number of possible configurations (Figure S4) indicate that both materials should cycle in a comparable voltage window and their capacity would not be limited by thermodynamics.”

To address the specific concerns of the reviewer: (a) First, the deviation of real compositions from the targeted ones is extremely small as confirmed by elemental analysis (within 1-1.5% of all cation sites for all elements from the designed composition). Second, at least 20% of Li excess is detected in both compositions which would guarantee good Li percolation if all cations were randomly distributed. (b) Yes surface phases may form in-situ during cycling of ordered cathode materials, such as a densified spinel-like or rocksalt-like phase on the surface of NMC cathodes, but there is simply no evidence here that they limit the kinetics in the initial cycles. We note that this is not an unusual position to take. All NCA and NMC materials form densified surface

phases after a few cycles, but people still make significant statements about the Li kinetics in these materials based on classic electrochemical measurements.

The pronounced bump from 17° to 25° in the diffraction pattern of LMZO is associated with the stronger short-range order in LMZO (consistent with Figure S9). A typical X-ray diffractometer hardly has the resolution to identify specific short-range order; nevertheless, such information is all contained within the pattern, although buried underneath the background, and shows up as humps in diffraction patterns. The extra peak at 28° in Figure S1 is from the high-vacuum grease (Dow Corning), as seen below in a separate XRD pattern of the grease only. We now update that **Figure S1** with the same data as used in Figure 1c, where no grease was used for sample preparation.

Figure caption: XRD pattern of high-vacuum grease (Dow Corning).

The difference in the voltage profiles in the high voltage region is likely due to the limited amount of extractable Li in LMZO as compared to LMTO. In LMZO, the plateau region does not show up before it exhausts all the kinetically extractable Li. The computed voltage profile of the LMZO by enumerating a large number of possible configurations (Figure S4) also shows that the plateau region appears after charging beyond $x = 0.6$, while that of LMTO appears earlier after $x = 0.4$.

Now that we understand the completely different cation short-range order in the two materials, it is to be expected that the SRO will also affect the voltage profile, which is a reflection of the Li-site energies in the lattice. It would be a very interesting research question for a future project to unravel the complex relationship of SRO and voltage. We have added a comment to the discussion section of the revised manuscript: “Furthermore, given that SRO differs significantly with compositions and fully controls the local environment around both Li and O ions, we expect that voltage profiles as well as the conditions for oxygen redox will also be modified by SRO.”

(3) Following by the previous question, the high voltage plateau for LMTO is quite similar to that of layered Li rich oxide with Li₂MnO₃ component, is there any possibility for the Li₂MnO₃-typed SRO? If not, what the origin for this high voltage plateau?

Answer: This voltage plateau was first observed in LMTO by Yabuuchi et al. and was attributed to oxygen oxidation/loss [Yabuuchi, Naoaki, et al. *Nature communications* 7 (2016): 13814.]. Oxygen activity is also the most common explanation for the voltage plateau in Li_2MnO_3 , where nearly all capacity comes from oxygen redox. A similar plateau has also been observed in $\text{Li}_{1.3}\text{Mn}_{0.4}\text{Nb}_{0.4}\text{O}_2$ where, once again, most capacity comes from oxygen oxidation/loss [Kan, Wang Hay, et al. *Chemistry of Materials* 30.5 (2018): 1655-1666.]. Since the voltage plateau is mainly indicative of the redox mechanism, it is unlikely that it has anything to do with SRO.

(4) Based on the presented results, the SRO will reduced the content of Li4 clusters that in favor of lithium diffusion. Is there possibility that this negative impact be counteracted by increasing the lithium content in pristine materials? Any experiment results to support?

Answer: In this work we intentionally do not vary the Li content to make a fair comparison between various metal species and to study how their local interactions lead to different cation short-range orders and therefore different levels of kinetically accessible Li. However, as suggested by the reviewer, increasing the Li content will definitely lead to an increasing population of Li-rich local environments and therefore an increasing amount of percolating Li, as demonstrated by previous computational and experimental work from our group [Lee, Jinhyuk, et al. *Science* 343.6170 (2014): 519-522; Urban, Alexander, Jinhyuk Lee, and Gerbrand Ceder. *Advanced Energy Materials* 4.13 (2014): 1400478.].

In terms of experimental evidence, we refer the referee to a previous work published by our group [Lee, Jinhyuk, et al. *Energy & Environmental Science* 8.11 (2015): 3255-3265.] where both reversible capacity and rate capability were significantly improved by increasing the Li-excess level in a class of Li-Ni-Ti-Mo oxides. We also want to point out that cation short-range order does not necessarily have a negative impact on Li percolation. In Figure 6b, we show that a few compositions (e.g. $\text{Ni}^{2+}\text{-Ti}^{4+}$, $\text{Ni}^{2+}\text{-Mo}^{6+}$) have a percolating Li level above 80%, which is even better than the random case (70%).

(5) The authors proposed some general rules for selecting high performance cation-disordered oxide with desired SRO as displayed in Figure 5, then the reviewer wishes to see the actual experimental data with the calculated most improved material.

Answer: The examples with the highest expected percolating Li content >80% is shown in Figure 6b (i.e. $\text{Ni}^{2+}\text{-Ti}^{4+}$ and $\text{Ni}^{2+}\text{-Mo}^{6+}$). These metal species have been demonstrated to have very good Li percolation in a previous publication [Lee, Jinhyuk, et al. *Energy & Environmental Science* 8.11 (2015): 3255-3265.], which shows a reversible capacity of 250 mAh/g (~0.83 Li/f.u.) for $\text{Li}_{1.2}\text{Ni}_{1/3}\text{Ti}_{1/3}\text{Mo}_{2/15}\text{O}_2$ when cycled at 10 mA/g between 1.5 – 4.5 V at room temperature.

We added the following sentences to the discussion about Figure 6b: “From Figure 6b, we also notice that some of the metal pairs with the highest predicted percolating Li levels, i.e. $\text{Ni}^{2+}\text{-Ti}^{4+}$ and $\text{Ni}^{2+}\text{-Mo}^{6+}$, have been experimentally realized in a compound $\text{Li}_{1.2}\text{Ni}_{1/3}\text{Ti}_{1/3}\text{Mo}_{2/15}\text{O}_2$ to deliver a high reversible capacity of 250 mAh/g when cycled at

10 mA between 1.5–4.5 V at room temperature [Lee, Jinhyuk, et al. Energy & Environmental Science 8.11 (2015): 3255-3265.]”

(6) Please note that the term “diffusion coefficient” labeled on Figure 2d should be “apparent diffusion coefficient”, which cannot be directly linked to the lithium diffusion between atomic sites.

Answer: We agree with the reviewer and have changed that Figure label to “apparent diffusion coefficient”.

Once again, we thank both reviewers for their extensive comments.

Reviewers' comments:

Reviewer #1 (Remarks to the Author):

Response to the Reviewers: Manuscript ID NCOMMS-18-26372-T

As a conclusion of my first review wrote "The thoughts brought into this manuscript are interesting; therefore the paper raised several questions and concerns that must be thoroughly addressed prior it can be considered for publication"

Looking at the rebuttal letter, I can state that the authors did thoroughly answer the questions and I appreciate.

- 1) All the doubts that I had about the coherence between RT and 55°C data are now clarified with the new plots, namely the normalization.
- 2) The issues of grain boundaries. No solution is provided owing to the complexity of the issue but I am satisfied with the addition paragraph that the authors are proposing.
- 3) Cluster expansion question: thanks for the clarification and addition.
- 4) Question 5, I still not fully agree but it is not critical
- 5) Question 6: Will be nice for the authors to use their last sentence which transmit a clear message "Cation short-range order is crucial in determining kinetically extractable Li in DRX materials as Li transport relies on percolating through a 3D network"

In short, I am satisfied with the answers and the clarity that the paper has gained so that I give my GREEN LIGHT for publication.

Reviewer #2 (Remarks to the Author):

I went through the revised manuscript carefully and feels that there are still several issues need to be clarified before this manuscript could be suitable for publication in Nature Communications.

1. While I have no doubt on the calculation and simulation results, and I also agree that short-range ordering (SRO) will affect the Li diffusion kinetics, it is insufficient to conclude that it's the major cause to the different electrochemical behavior between LMZO and LMTO as presented..

(1) The authors updated a new set of electrochemical data (Figure R2 and Figure 2c), however, this new results are not well consistent with the previously presented data set (Figure R1 and Figure 2a in the old version). The new data of LMZO shows obviously higher capacity and clearer voltage plateau at high voltage than the previous one at 50°C and at charging cut-off voltage of 4.5V, that means the electrochemical behavior is very sensitive to other kinetic issues, not only the bulk Li⁺ diffusivity in the material.

(2) The estimation of the Li⁺ diffusivity is based on the calculation and simulation of the initial structure (as-synthesized material), indicating a significant Li⁺ diffusion behavior between LMZO and LMTO. However, the apparent diffusion coefficient obtained from the experiment measurement shows similar results for LMZO and LMTO at initial states (Figure 2f), the authors may need to explain and comment on it in the main text.

Overall, the authors need to provide more solid experimental proofs to support the calculation results. To make a complete story, the authors may need either provide additional experimental results, such as NMR or AC impedance, as another reviewer also suggested, or have a more general discussions.

2. It is great to see that the authors proposed some general rules for selecting high performance

cation disordered oxide with desired SRO, however, whether it is valid or at what circumstances it is valid, needs more solid experimental results to support or more in general discussions.

For example, the results in Figure 6b and also the author's statements in rebuttal letter (expanded as follow) indicates that "Ni²⁺-Ti⁴⁺" might be a good combination for Li percolation. And the authors referred to reviewer to read their previous paper (EES, 2015, 8, 3255) and claimed that "Ni-Ti-Mo" system shows high capacity and good reversibility. However, it is "Ni-Ti" binary system, not "Ni-Ti-Mo" ternary system, the theory predicted to have better kinetics in the current manuscript. In fact the Li_{1.2}Ni_{0.4}Ti_{0.4}O₂ shows really poor performances with ~ 110 mAh/g (Figure 4 and Figure 5 in EES paper)

I would not say that these results are in contradiction to current theoretic predictions, but the authors may definitely need to comment on it..

Authors' response to Reviewer 2's Question 5 in rebuttal letter:

The examples with the highest expected percolating Li content >80% is shown in Figure 6b (i.e. Ni²⁺-Ti⁴⁺ and Ni²⁺-Mo⁶⁺). These metal species have been demonstrated to have very good Li percolation in a previous publication [Lee, Jinhyuk, et al. Energy & Environmental Science 8.11 (2015): 3255-3265.], which shows a reversible capacity of 250 mAh/g (~0.83 Li/f.u.) for Li_{1.2}Ni_{1/3}Ti_{1/3}Mo_{2/15}O₂ when cycled at 10 mA/g between 1.5–4.5 V at room temperature.

Response to the Reviewers: Manuscript ID NCOMMS-18-26372A

We again thank the editor and both reviewers for their time spent to evaluate our manuscript and for their recommendation/comments. All of the reviewers' concerns are addressed below in detail. All changes made in the manuscript are highlighted in blue.

Response to Reviewer #1

As a conclusion of my first review wrote "The thoughts brought into this manuscript are interesting; therefore the paper raised several questions and concerns that must be thoroughly addressed prior it can be considered for publication"

Looking at the rebuttal letter, I can state that the authors did thoroughly answer the questions and I appreciate.

1) All the doubts that I had about the coherence between RT and 55°C data are now clarified with the new plots, namely the normalization.

2) The issues of grain boundaries. No solution is provided owing to the complexity of the issue but I am satisfied with the addition paragraph that the authors are proposing.

3) Cluster expansion question: thanks for the clarification and addition.

4) Question 5, I still not fully agree but it is not critical

5) Question 6: Will be nice for the authors to use their last sentence which transmit a clear message "Cation short-range order is crucial in determining kinetically extractable Li in DRX materials as Li transport relies on percolating through a 3D network"

In short, I am satisfied with the answers and the clarity that the paper has gained so that I give my GREEN LIGHT for publication.

Response: We thank Reviewer #1 for his/her recommendation for publication. Based on the reviewer's new comment 5), we added the following sentence to the end of the second paragraph in Introduction: "We find that cation short-range order (SRO) is important in determining the amount of kinetically extractable Li in DRX materials as the Li transport relies on percolation through a 3D network."

Response to Reviewer #2

I went through the revised manuscript carefully and feels that there are still several issues need to be clarified before this manuscript could be suitable for publication in Nature Communications.

We appreciate Reviewer #2's comments and thoughtful questions for improving the overall clarity of the manuscript.

1. While I have no doubt on the calculation and simulation results, and I also agree that short-range ordering (SRO) will affect the Li diffusion kinetics, it is insufficient to conclude that it's the major cause to the different electrochemical behavior between LMZO and LMTO as presented..

(1) The authors updated a new set of electrochemical data (Figure R2 and Figure 2c), however, this new results are not well consistent with the previously presented data set (Figure R1 and Figure 2a in the old version). The new data of LMZO shows obviously higher capacity and clearer voltage plateau at high voltage than the previous one at 50oC and at charging cut-off voltage of 4.5V, that means the electrochemical behavior is very sensitive to other kinetic issues, not only the bulk Li+ diffusivity in the material.

(2) The estimation of the Li+ diffusivity is based on the calculation and simulation of the initial structure (as-synthesized material), indicating a significant Li+ diffusion behavior between LMZO and LMTO. However, the apparent diffusion coefficient obtained from the experiment measurement shows similar results for LMZO and LMTO at initial states (Figure 2f), the authors may need to explain and comment on it in the main text.

Overall, the authors need to provide more solid experimental proofs to support the calculation results. To make a complete story, the authors may need either provide additional experimental results, such as NMR or AC impedance, as another reviewer also suggested, or have a more general discussions.

Answer: We thank reviewer #2 for the further evaluation of our work.

- (1) We agree with the reviewer that the data in Figure R1 and R2 is slightly different. There are two differences. (i) The discharge capacity in Figure R2 is slightly higher, which we attribute to the lower charge voltage for R2 (4.5V) than for R1 (4.7V). The high charge voltage in R1 likely causes some surface decomposition reaction, as evidenced by an extra plateau above 4.5 V at 50 °C, leading to impedance growth and a lower discharge voltage and capacity. The overpotential difference during discharge can be observed when plotting all curves on the same graph (Figure R4). (ii) In addition, the plateau in R2 at ~4.1 V during charge is somewhat more pronounced than in R1. We do not know why this is the case and believe that it may be due to sample variations or variations in testing conditions. We have however repeated the test in R2 with several samples (Figures R5 and R6) and always get the result shown in R2. More importantly, because the extra plateau above 4.5 V at 50°C may involve surface decomposition reactions and oxygen oxidation we have restricted our analysis of the kinetics to the Li transport below the plateau voltage. This was our intent with showing the new data.

Figure R4. Discharge voltage profiles of LMZO after being charged to 4.7 V (red curve) or 4.5 V (black curve) at 10 mA/g at 50°C.

Figure R5. Voltage profiles of a new cell of LMZO between 1.5 – 4.5 V at 50°C at 10 mA/g.

Figure R6 Voltage profiles of a second new cell of LMZO between 1.5 – 4.5 V at 50°C at 10 mA/g.

(2) With regard to the comment of the reviewer that the initial diffusivity of LMZO and LMTO is similar, we would like to clarify our previous response: PITT measures the chemical diffusivity and this is what is shown in Figure 2f. The chemical diffusivity, D_{chem} , is the product of the self-diffusivity, D_{self} , and the thermodynamic factor: $D_{\text{chem}} = F D_{\text{self}}$. The self-diffusivity reflects the intrinsic mobility of Li^+ while the thermodynamic factor includes the driving force provided by a concentration gradient. For an intercalation cathode F is formally equal to dV/dx where V is voltage and x is the amount of Li removed. Because of the steep initial slope of the voltage curve, F is very large and dwarfs any variation in intrinsic diffusivity for $x < 0.05$. However, it is clear that for slightly larger x , where the thermodynamic factor is smaller, the diffusivity of Li in LMTO is much higher than in LMZO. This is therefore clear evidence that the intrinsic Li diffusivity in LMTO is higher than in LMZO. We thus added the following discussion following the sentence “would only impact the bulk diffusivity” in the description of the PITT results: “Given that the chemical diffusivity measured by PITT is a product of the thermodynamic factor and the self-diffusivity, the apparent diffusivities (D_{Li} 's) of the two materials at the beginning of charge ($x < 0.05$) are dominated by the large thermodynamic factors originating from the steep voltage increase in the region and are therefore not a representation of the intrinsic Li mobility.”

2. It is great to see that the authors proposed some general rules for selecting high performance cation disordered oxide with desired SRO, however, whether it is valid or at what circumstances it is valid, needs more solid experimental results to support or more in general discussions.

For example, the results in Figure 6b and also the author's statements in rebuttal letter (expanded as follow) indicates that “Ni²⁺-Ti⁴⁺” might be a good combination for Li percolation. And the authors referred to reviewer to read their previous paper (EES, 2015, 8, 3255) and claimed that “Ni-Ti-Mo” system shows high capacity and good reversibility. However, it is “Ni-Ti” binary system, not “Ni-Ti-Mo” ternary system, the theory predicted to have better kinetics in the current manuscript. In fact the Li_{1.2}Ni_{0.4}Ti_{0.4}O₂ shows really poor performances with ~ 110 mAh/g (Figure 4 and Figure 5 in EES paper)

I would not say that these results are in contradiction to current theoretic predictions, but the

authors may definitely need to comment on it..

Authors' response to Reviewer 2's Question 5 in rebuttal letter:

The examples with the highest expected percolating Li content >80% is shown in Figure 6b (i.e. Ni²⁺-Ti⁴⁺ and Ni²⁺-Mo⁶⁺). These metal species have been demonstrated to have very good Li percolation in a previous publication [Lee, Jinhyuk, et al. Energy & Environmental Science 8.11 (2015): 3255-3265.], which shows a reversible capacity of 250 mAh/g (~0.83 Li/f.u.) for Li_{1.2}Ni_{1/3}Ti_{1/3}Mo_{2/15}O₂ when cycled at 10 mA/g between 1.5–4.5 V at room temperature.

Answer: We appreciate Reviewer #2's comments but we believe there is some confusion. The reviewer refers to a composition Li_{1.2}Ni_{0.4}Ti_{0.4}O₂ in the EES paper by Lee, and refers to its performance data in Figure 4 and 5 of that paper. However, the only composition without Mo considered in that EES paper is LiNi_{0.5}Ti_{0.5}O₂, **not** Li_{1.2}Ni_{0.4}Ti_{0.4}O₂. The performance of LiNi_{0.5}Ti_{0.5}O₂ in that paper is poor because the Li content is below the percolation limit. The composition Li_{1.2}Ni_{0.2}Ti_{0.6}O₂, listed as one of the predictions in our work in Figure 6, has the right Ni (2+) and Ti (4+) valence states and favorable SRO but offers only a small theoretical TM capacity because of the low Ni content. To further clarify this issue and discuss the conditions under which our predictions are valid, we added the following paragraph to the Discussion section following the sentence "between 1.5–4.5 V at room temperature [ref]": "However, it is worth noting that favorable cation SRO and a high kinetically accessible Li content only serve as necessary conditions and yet are insufficient to guarantee good capacity, which relies on various other factors including the TM capacity, electronic conductivity, particle size, etc. For instance, although we predict a high kinetically accessible Li content >80% for Li_{1.2}Ni_{0.2}Ti_{0.6}O₂ (Figure 6b), this compound is unlikely to deliver a high reversible capacity given its limited Ni content and the resulting small theoretical TM capacity."

The confusion may have arisen from the fact that in the original draft we did not include information about the Li excess levels of the compositions shown in Figure 6. We now include that information in the caption of Figure 6 by adding: "All the compositions listed have the same Li excess level of 20%."

REVIEWERS' COMMENTS:

Reviewer #2 (Remarks to the Author):

The authors addressed all issues properly and the revised manuscript is now suitable to be published in Nature Communications.

Response to the Reviewers

We thank both reviewers for their time spent to evaluate our manuscript and their recommendation for publication. Below is the final referee report. Given that no questions are raised, we do not make further changes in the manuscript.

=====

REVIEWERS' COMMENTS:

Reviewer #2 (Remarks to the Author):

The authors addressed all issues properly and the revised manuscript is now suitable to be published in Nature Communications.